**Data Availability Statement:** All relevant data are within the manuscript and its Supporting information files.

**Funding:** The author(s) received no specific funding for this work.

# A novel methionine nanoparticle in broiler chickens: Bioavailability and requirements

**Mahmoud Ghazaghi[1], Mehravar Mehri[2], Morteza Asghari-Moghadam[1], Mehran Mehri [1]***

**1** Department of Animal Sciences, Faculty of Agriculture, University of Zabol, Zabol, Sistan, IRAN,
**2** Department of Education, Shahid Motahari High School, Ministry of Education, Zahedan, IRAN

* mehri@uoz.ac.ir

## Abstract

This bioassay evaluated the bioavailability (RBV) of a novel nanoparticle of methionine (nano-Met) relative to $_{DL}$-methionine (DL-Met), and estimated methionine requirements for both sources in starting broilers. Five supplemental levels (0.05, 0.10, 0.15, 0.20, and 0.25% of diet) of DL-Met or nano-Met were added to a basal diet containing 0.35% standardized ileal digestible (SID) methionine to create 11 experimental diets, including a basal diet and 10 experimental diets containing 0.40, 0.45, 0.50, 0.55, and 0.60% SID-Met, respectively. A total of 825 one-day-old male Ross 308 birds were randomly assigned to 11 treatments with 5 pen replicates and 15 birds each. Body weight gain (BWG), breast meat yield (BMY), and thigh meat yield (TMY) increased ($P < 0.001$) while feed conversion ratio (FCR) and malondialdehyde (MDA) concentration in meat samples decreased ($P < 0.001$) with increasing dietary methionine. Based on the slope-ratio method, the RBV of nano-Met relative to DL-Met for BWG, FCR, and TMY were 102 (48–155%; $R^2 = 0.71$), 134 (68–201%; $R^2 = 0.77$), and 110% (27–193%; $R^2 = 0.55$), respectively. Considering the statistical accuracy of the spline models, the estimated values of DL-Met for maximum BWG and nano-Met for maximum TMY were 0.578% and 0.561%, respectively, which were statistically higher than those recommended for commercial settings. The highest effect size of supplemental methionine was on MDA ($\eta^2p = 0.924$), followed by FCR ($\eta^2p = 0.578$), BMY ($^2p = 0.575$), BWG ($\eta^2p = 0.430$), and TMY ($\eta^2p = 0.332$), suggesting the potent antioxidant properties of methionine. Our findings suggest that reducing the particle size of DL-Met to nanoparticles could be a promising strategy to enhance the efficiency of methionine supplementation in broilers, an idea that requires further investigation in future research.

## Introduction

Methionine (Met) is an essential amino acid that plays a crucial role in the growth and development of broiler chickens. It is the first limiting amino acid in corn and soybean meal-based broiler diets, and its uptake from the diet is required for optimal growth and development [1, 2]. However, the bioavailability of methionine can be influenced by various factors, such as the source (DL-Met *vs*. 2-hydroxy-4-(methylthio) butanoic acid) and form of methionine, as well as the presence of other amino acids in the diet [3]. Therefore, it is important to determine the

**Competing interests:** The authors have declared that no competing interests exist.

bioavailability of methionine in broiler diets to ensure that the dietary requirements are being met.

This has been a topic of research in animal studies, with experiments conducted to determine the relative bioavailability of different forms of methionine [4–7]. In addition, there is a growing interest in the impact of methionine bioavailability on broiler performance, with studies exploring the effects of methionine on intestinal development, immune response, and antioxidant system in broilers [6]. Biological availability (bioavailability) refers to the amount of a nutrient that can be absorbed and utilized by the body [8, 9]. Factors that influence methionine bioavailability include source, form, presence of other ingredients in the diet, and metabolic processes in the gut. Poor dietary quality and lack of balance in the diet can lead to low methionine availability, making it important to address poultry nutrition. In this context, several studies have shown that different sources of methionine, such as HMTBA, exhibited lower bioavailability than DL-Met for broilers and laying hens [10]. However, Kratzer and Littell [11] suggested that separate plateau models should be used when comparing these two products (DL-Met and HMTBA) and that forcing a common plateau resulted in model bias.

Nanotechnology is a rapidly growing field that has the potential to revolutionize various industries, including agriculture. In recent years, there has been a growing interest in the application of nanotechnology in broiler production, with studies exploring the potential benefits of nanotechnology-based tools for improving broiler performance and health [12, 13]. Nanotechnology involves the precise manipulation and application of materials at a nanoscale, which is defined as 1 to 100 nanometers [14]. This allows for the creation of structures and systems that can be used in diverse aspects of broiler production, such as feed additives, vaccines, and antimicrobial agents [15]. The use of nanotechnology in broiler production has the potential to improve feed efficiency, enhance immune response, and reduce the risk of disease transmission [13]. In poultry nutrition, nanoparticles can be used to enhance the availability and absorption of nutrients, such as vitamins and minerals, as well as to deliver nutrients more effectively to targeted organs or cells. Nanoparticles can also be used as carriers for vaccines, drugs, and other therapeutic agents [16]. In the field of poultry nutrition, nanoparticles are being increasingly studied as a tool to enhance the efficiency and effectiveness of nutrient delivery. For example, nanoparticles can be used to increase the solubility of fat-soluble vitamins, such as vitamin E, and to improve their absorption and utilization in poultry [17]. Additionally, nanoparticles can be designed to target specific organs or cells and to extend the time that nutrients remain in the gut, leading to improved absorption and utilization.

In addition to the bioavailability of methionine, it is also crucial to accurately estimate the nutritional requirements for its successful implementation in diet formulation. This is particularly crucial to maximizing profits in poultry production, a process that can be achieved through the implementation of comprehensive regression modeling techniques such as spline models.

In dose-response studies, different nutrient levels are tested to determine the nutrient requirement. It is important to allocate most of the levels around an expected requirement and to include one level associated with the best performance. The level associated with the best performance is determined through statistical analysis, which is crucial to accurately determine the nutrient requirement [18]. The most appropriate approach to determine a nutritional requirement is to model a response curve using linear and non-linear models. The use of different models for estimation is important because it allows for precise optimization of responses without moving into inhibiting or toxic ranges of nutrients [18]. The models may be used to estimate nutritional requirements or the most economical feeding levels of critical nutrients [19]. Broken-line models such as one-slope and two-slope models may provide different optimal amounts of nutrients. The choice of the best model could be derived from

selecting the one with the highest accuracy and the lowest error [20]. The application of various models allows for selecting the most accurate estimate from a range of calculations for a single nutrient in dose-response studies [21].

The objective of this research is to introduce an innovative methionine supplement utilizing nanosized particles. Additionally, the study aims to assess the relative bioavailability (RBV) of methionine nanoparticles (nano-Met) compared to DL-Met in early-stage broilers. Furthermore, it seeks to determine the methionine requirements for two different sources of methionine by evaluating performance metrics and utilizing oxidation markers in meat samples.

## Materials and methods

### Ethics statement

The committee responsible for animal ethics at the University of Zabol and the Iranian Council of Animal Care approved this experimental protocol. The study followed the "Animal Research: Reporting of In Vivo Experiments" (ARRIVE) guidelines (https://arriveguidelines.org) and the National Institutes of Health Guidelines for the Care and Use of Laboratory Animals.

### Bird management

A total of 825 one-day-old male broiler chicks (Ross 308) were randomly allotted to 11 treatments with 5 replicates and 15 birds each. Throughout the study period of 14 days, the broilers were housed in floor pens and received *ad libitum* access to mesh feed and water. The temperature and lighting were set to simulate commercial farm operations. Feed intake (FI), body weight gain (BWG), and feed conversion ratio (FCR) from 75 birds per treatment were measured. At the end of the experiment, 4 birds in each pen (20 birds per treatment) were euthanized using $CO_2$ asphyxiation, and breast meat yield (BMY) and thigh meat yield (TMY) were determined as percentage of live weight.

### Experimental diets

Basal diet: The corn–soybean-meal-based starter diet (Table 1) was limited in methionine (0.35% of diet) + Cys (0.35% of diet) but adequate in all other nutrients and energy [22].

Dose-response diets: Ten experimental diets were prepared with the addition of 5 supplemental levels (0.05, 0.10, 0.15, 0.20, and 0.25% of diet) of DL-Met or nano-Met to the basal diet at the expense of cornstarch to create 11 experimental diets including a basal diet and 10 experimental diets containing 0.40, 0.45, 0.50, 0.55, and 0.60% SID-Met, respectively.

All protein-containing feed ingredients were analyzed for their crude protein content and amino acid profiles, according to standard analytical methods [23]. The samples were subjected to 24-hour hydrolysis in 6 N hydrochloric acid, followed by oxidation of methionine and cysteine using performic acid. For analysis of tryptophan content, the samples were subjected to hydrolysis using barium hydroxide.

### Malondialdehyde assay

In this study, we used the third-order derivative spectrophotometry method to determine the malondialdehyde (MDA) concentration in meat samples as described by Botsoglou, Fletouris [24]. At the end of the experiment, 4 birds in each pen (20 birds per treatment) were euthanized and deboned breast meat samples were collected. One gram of grounded meat sample was picked up and homogenized (Polytron homogenizer, PCU, Switzerland) with 4 ml of 5% aqueous trichloroacetic acid (TCA) and 2.5 ml of 0.8% butylated hydroxytoluene, and then

**Table 1. Composition of basal diet.**

| Ingredient | Amount (g/kg) |
|---|---|
| Corn, Grain | 567.5 |
| Soybean Meal-44 | 309.4 |
| Corn Gluten Meal | 61.0 |
| Dicalcium Phosphate | 14.9 |
| Oyster Shells | 14.0 |
| Corn Starch | 10.0 |
| Sunflower Oil | 9.20 |
| Sodium Bicarbonate | 5.00 |
| L-Lysine HCl | 2.80 |
| Mineral Premix[1] | 2.50 |
| Vitamin Premix[2] | 2.50 |
| L-Thr | 1.10 |
| NaCl | 0.10 |
| Nutrient specifications | |
| Metabolizable energy (MJ/kg)[3] | 2950 |
| Crude protein (g/kg)[4] | 226 |
| Calcium (g/kg)[3] | 9.50 |
| Available phosphorus (g/kg)[4] | 4.50 |
| SID Met (g/kg)[4] | 3.50 |
| SID Cys (g/kg)[4] | 3.50 |
| SID Met + Cys (g/kg)[4] | 7.00 |
| SID Lys (g/kg)[4] | 11.9 |
| SID Arg (g/kg)[4] | 12.9 |
| SID Thr (g/kg)[4] | 8.30 |
| SID Trp (g/kg)[4] | 2.20 |
| SID Val (g/kg)[4] | 9.50 |
| DEB (mEq/kg)[5] | 250 |

[1]Mineral premix provided per kilogram of diet: Mn (from $MnSO_4 \cdot H_2O$), 65 mg; Zn (from ZnO), 55 mg; Fe (from $FeSO_4 \cdot 7H_2O$), 50 mg; Cu (from $CuSO_4 \cdot 5H_2O$), 8 mg; I [from Ca $(IO_3)_2 \cdot H_2O$], 1.8 mg; Se, 0.30 mg; Co (from $Co_2O_3$), 0.20 mg; Mo, 0.16 mg.

[2]Vitamin premix provided per kilogram of diet: vitamin A (from vitamin A acetate), 11,500 U; cholecalciferol, 2100 U; vitamin E (from dl-α-tocopheryl acetate), 22 U; vitamin $B_{12}$, 0.60 mg; riboflavin, 4.4 mg; nicotinamide, 40 mg; calcium pantothenate, 35 mg; menadione (from menadione dimethyl-pyrimidinol), 1.50 mg; folic acid, 0.80 mg; thiamine, 3 mg; pyridoxine, 10 mg; biotin, 1 mg; choline chloride, 560 mg; ethoxyquin, 125 mg.

[3]Calculated values.

[4]Analyzed values.

[5]DEB: dietary electrolyte balance represents dietary Na + K—Cl in mEq/kg of diet.

centrifuged at $3000 \times g$ for 3 min. The top layer (hexane) was discarded, and the bottom layer was filtered and made to 5 mL volume with 5% TCA, then placed into a screw-capped tube containing 3 mL of 0.8% aqueous 2-thiobarbituric acid (TBA). The tubes were heated at 70°C in a water bath for 30 min. After that, the tubes were cooled with tap water and then submitted to a spectrophotometer (UNIKON 933, Kontron Co. Ltd., Milan, Italy). The height of the third-order derivative peak that appeared at 521.5 nm was used to calculate the MDA concentration (expressed as milligrams per kilogram) as the secondary product of oxidation in the samples. The precursor of MDA in the standard curve was the tetraethoxypropane (1, 1, 3, 3-tetraethoxy propane, T9889, 97%, Sigma, USA).

### Preparation of methionine nanoparticles

The ultrasonic technique was used to produce methionine nanoparticles [25]. The preparation of methionine nanoparticles involved dissolving a specific amount of methionine in double-distilled water in a beaker on a magnetic stirrer. Ultrasonic waves were then applied to the beaker, and a specific amount of surfactant was added to prevent the nanoparticles from sticking together. Microwave radiation and ultrasonic waves with specific wattage and alternating cycles of percentage per minute were subsequently applied. The solution was then placed in a centrifuge, and the nanoparticles were separated from the liquid. The sediment from the centrifugation was subsequently dried in a vacuum oven, and the resulting product was registered as a patent (NO: 8463/07/10/1397).

### Scanning electron microscopy (SEM)

Scanning electron microscopy (SEM) is a highly versatile technique used in nanotechnology to obtain high-resolution images and detailed surface information of samples. SEM is used for many types of work in material science research, development, and quality control. We used this technique to observe the morphology of methionine nanoparticles (Fig 1a).

### Fourier transform infrared spectroscopy

A Fourier transform infrared spectrophotometer (FT-IR, Bruker, Germany) was used to investigate the interaction of functional groups between DL-methionine and nanoparticles of methionine (Fig 1b).

### Statistical analysis

Data were analyzed as a randomized complete design using the GLM procedure of SAS [26], and the mean of each pen was used as the experimental unit. Orthogonal-polynomial contrast

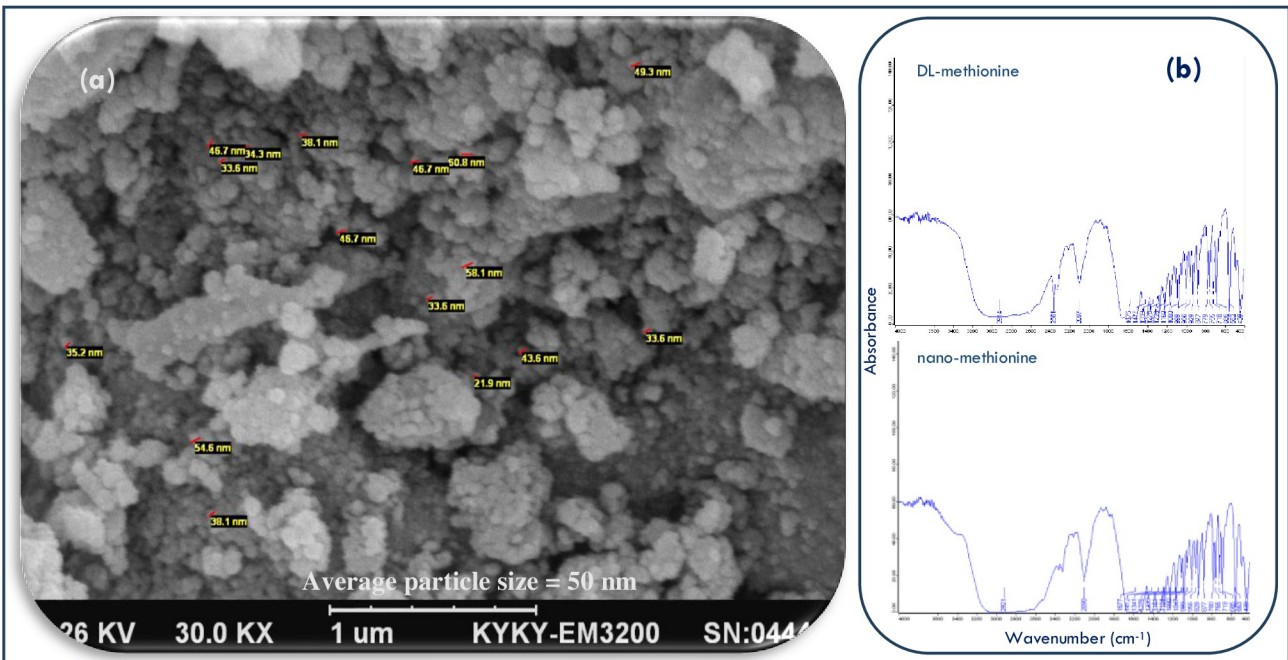

**Fig 1.** Scanning electron microscope (SEM) of methionine nanoparticles (a); Fourier transform infrared spectroscopy (b).

was also applied to determine the linear and quadratic effects of increasing levels of DL-Met or nano-Met on the response criteria (i.e., BWG, FCR, BMY, TMY, and MDA). To determine the methionine requirements different models of broken line regression were fitted according to Khosravi, Mehri [20] as follows:

One-slope broken line:

*Linear ascending (or descending)*:

$$Y = L + U \times (R - X) \times (X < R)$$

*Quadratic ascending (or descending)*:

$$Y = L + U \times (R - X)^2 \times (X < R)$$

Two-slope broken line:

*Linear ascending (or descending)-linear descending (or ascending)*:

$$Y = L + U \times (R - X) \times (X < R) + V \times (X - R) \times (X > R)$$

*Quadratic ascending (or descending)-linear descending (or ascending)*:

$$Y = L + U \times (R - X)^2 \times (X < R) + V \times (X - R) \times (X > R)$$

*Quadratic ascending (or descending)-quadratic descending (or ascending)*:

$$Y = L + U \times (R - X)^2 \times (X < R) + V \times (X - R)^2 \times (X > R)$$

where Y is the bird response; L is the asymptote for the first segment; U and V are the slopes for the first and second lines, respectively, with increasing or descending slope, and R is the break point that is considered as a "*requirement*" point. The best estimation of the optimal amount of dietary DL-Met or nano-Met for each response was chosen based on the highest $R^2$ and lowest root mean squared error (RMSE).

Common-intercept, multiple linear regression, and slope-ratio assay [27] were used to estimate the bioavailability of nano-Met. Statistical and fundamental validity of the dose-response assay was achieved by utilizing analysis of variance (ANOVA) to test for linearity of slopes, lack of curvature, and equality of intersection of reference and test diets at the basal response, as suggested by Finney [27]. The BWG, FCR, and TMY were considered the bird response criteria, and bioefficacy values were determined by regressing bird responses versus supplemental dietary concentrations of the methionine sources [4]. The pen mean was used as the experimental unit for all statistical analyses. The NLIN procedure in SAS/STAT software (2002) was applied to fit linear models based on the slope-ratio method as follows:

$$Y = a + b_1 x_1 + b_2 x_2$$

where Y = bird response (BWG, FCR, and TMY), a = intercept (bird response with basal diet), b = asymptotic response, a + b = common asymptote (maximum response level), $b_1$ = steepness coefficient for DL-Met, $b_2$ = steepness coefficient for nano-Met, and $x_1$, $x_2$ = dietary level of DL-Met, nano-Met, respectively, and e = the random error. According to the procedure outlined by Littell, Henry [28], the bioavailability of nano-Met compared with DL-Met was given by $b_2/b_1$, the ratio of regression coefficients.

## Results

### Scanning of nanoparticles

The scanning electron microscope was used to obtain essential information about the detection, morphology, and particle size of the nanoparticles, as depicted in Fig 1a. Additionally, the infrared (IR) spectrum, illustrated in Fig 2b, provided insights into the functional groups present in the sample, enabling the identification of chemical bonds and confirmation of nanoparticle formation. A comparison of the spectra of the substrate (DL-Met) and the product (nano-Met) confirmed that the methionine nanoparticles possess the functional groups of the substrate, thus validating the synthesis process.

### Performance and MDA

As indicated by the performance data (Table 2), neither supplemental levels of methionine nor methionine sources affected FI. Body weight gain ($P < 0.001$), BMY ($P < 0.001$), and TMY ($P = 0.010$) increased with increasing supplemental methionine, while FCR and MDA decreased ($P < 0.001$). Birds fed the basal diet deficient in Met achieved the lowest BWG (28.02 g/b), BMY (17.93%), and TMY (16.38%) while the highest FCR (1.69) and MDA (3.867 mg/kg). The highest effect size of supplemental methionine was on MDA ($\eta^2 p = 0.924$), followed by FCR ($\eta^2 p = 0.578$), BMY ($\eta^2 p = 0.575$), BWG ($\eta^2 p = 0.430$), and TMY ($\eta^2 p = 0.332$). The effect size of the methionine source was highest on carcass attributes (BMY: $\eta^2 p = 0.355$; TMY: $\eta^2 p = 0.210$) and its effect was lowest on FCR ($\eta^2 p = 0.001$).

### Statistical validation for bioavailability assay

To determine the bioavailability of a nutrient, the slope-ratio method was employed and its validity was tested using ANOVA, as per Finney [27]. The statistical validity was achieved by checking for linearity of responses to the reference and test diets, while the fundamental validity was assessed by determining whether regression lines for these diets intersect at the point of the basal diet. The present experiment indicated that the assumptions regarding linearity, lack of curvature, and intersection were valid for BWG, FCR, and TMY, which were the dependent variables, and supplemental levels of methionine, which were the associated independent variables (Table 3). The assumption for statistical validity and fundamental validity did not hold for the dependent variables BMY and MDA.

### Estimation of methionine bioavailability

Based on multiple linear regression and slope-ratio method, the RBV of nano-Met for BWG, FCR, and TMY were estimated at 102, 134, and 110% (Fig 2). However, the range of confidence interval for the RBV estimates showed that the differences between test and standard diets were not significant. Based on the slope-ratio method, the RBV of nano-Met relative to DL-Met for BWG, FCR, and TMY were 102 (48–155%; $R^2 = 0.71$), 134 (68–201%; $R^2 = 0.77$), and 110% (27–193%; $R^2 = 0.55$), respectively.

### Methionine requirements for BWG

Methionine requirements of DL-Met and nano-Met for maximum BWG were estimated at 0.488 and 0.500, 0.578 and 0.550, and 0.476 and 0.530% of diet using different regression models (Table 4; Fig 3). One-slope broken line with quadratic ascending portion (QBL: $R^2 = 0.937$, RMSE = 0.243) and two-slope linear-ascending linear descending (LALD: $R^2 = 0.896$, RMSE = 0.312) gave the best estimations, 0.578 and 0.530% for DL-Met and nano-Met, respectively, based on the highest $R^2$ and lowest RMSE. Considering confidence intervals (CIs), the

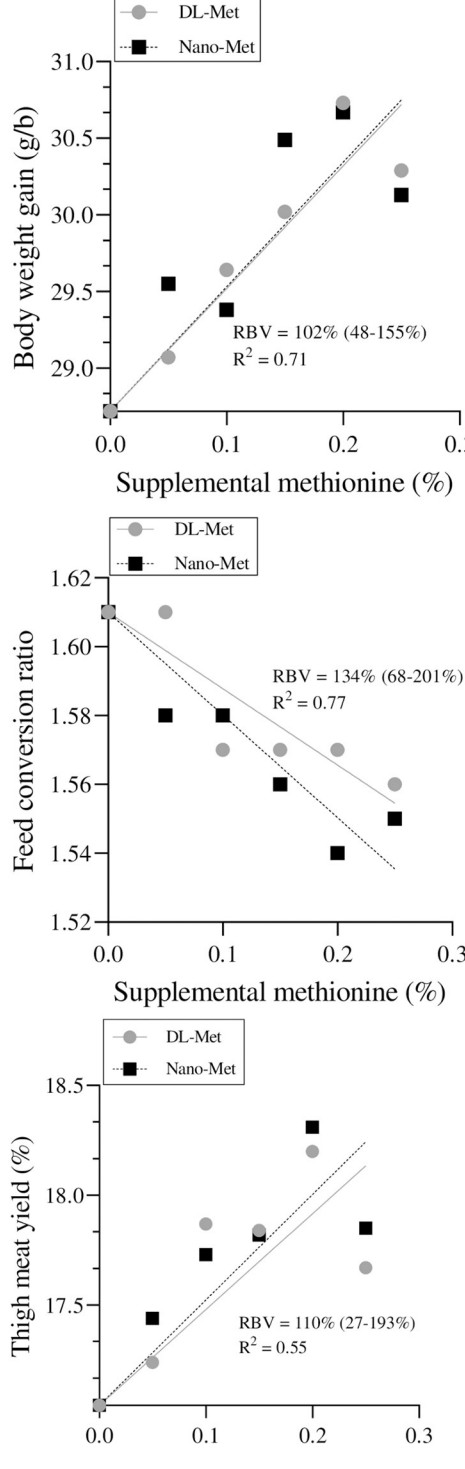

**Fig 2. The relative bioavailability of methionine nanoparticles (nano-Met) to DL-methionine for body weight gain, feed conversion ratio, and thigh meat yield using slope-ratio method.**

**Table 2. Effects of dietary DL-Met or nano-Met supplementation on feed intake (FI, g/b), body weight gain (BWG, g/b), feed conversion ratio (FCR), breast meat yield (BMY, %), thigh meat yield (TMY, %), and malondialdehyde (MDA, mg/kg).**

| Response | Basal diet | Experimental diets | | | | | | | | | | | Probability | | | | | SEM | Power |
| --- | --- | --- | --- | --- | --- | --- | --- | --- | --- | --- | --- | --- | --- | --- | --- | --- | --- | --- | --- |
| | | DL-Met | | | | | nano-Met | | | | | | Met | Source | Met × Source | Linear | Quadratic | | |
| | | 0.05 | 0.1 | 0.15 | 0.2 | 0.25 | 0.05 | 0.1 | 0.15 | 0.2 | 0.25 | | | | | | | | |
| FI | 47.2 | 46.7 | 46.5 | 47.2 | 48.2 | 47.1 | 46.7 | 46.5 | 47.6 | 47.1 | 46.6 | | 0.473 | 0.59 | 0.915 | 0.602 | 0.974 | 0.64 | 0.95 |
| BWG | 28.02 | 29.07 | 29.64 | 30.02 | 30.73 | 30.29 | 29.55 | 29.38 | 30.49 | 30.67 | 30.13 | | <0.001 | 0.815 | 0.151 | <0.001 | 0.042 | 0.58 | 0.97 |
| FCR | 1.63 | 1.61 | 1.57 | 1.57 | 1.57 | 1.56 | 1.58 | 1.58 | 1.56 | 1.54 | 1.55 | | <0.001 | 0.312 | 0.828 | <0.001 | 0.009 | 0.02 | 0.95 |
| BMY | 17.93 | 18.86 | 19.71 | 19.92 | 21.34 | 21.43 | 18.54 | 20.53 | 20.38 | 21.42 | 21.87 | | <0.001 | <0.001 | 0.062 | <0.001 | 0.01 | 0.58 | 0.96 |
| TMY | 16.38 | 17.24 | 17.87 | 17.84 | 18.2 | 17.67 | 17.44 | 17.73 | 17.82 | 18.31 | 17.85 | | 0.01 | 0.004 | 0.277 | 0.009 | 0.388 | 0.43 | 0.95 |
| MDA | 3.867 | 2.538 | 1.44 | 0.825 | 0.475 | 0.453 | 1.823 | 1.61 | 0.46 | 0.418 | 0.368 | | <0.001 | 0.145 | 0.323 | <0.001 | <0.001 | 0.20 | 0.98 |

**Table 3. Statistical validity of the bird responses for the bioavailability analysis.**

| Item | Probability (α = 0.05) | | | | |
|---|---|---|---|---|---|
| | **G** | **FCR** | **BMY** | **TMY** | **MDA** |
| Average slope | 0.001 | 0.001 | 0.001 | 0.05 | 0.001 |
| Slope difference | 0.95 | 0.29 | 0.001 | 0.01 | 0.88 |
| Blank | 0.13 | 0.20 | **0.01** | 0.13 | **0.001** |
| Intersection | 0.62 | 0.67 | **0.001** | 0.08 | 0.11 |
| Curvature | 0.75 | 0.88 | 0.41 | 0.13 | 0.001 |

G: body weight gain; FCR: feed conversion ratio; BMY: breast meat yield; TMY: thigh meat yield, MDA: malondialdehyde. The highlighted figures revealed that the model did not meet the statistical requirements for validity.

**Table 4. The best models were chosen on the basis of the maximum $R^2$ and the minimum root mean squared error (RMSE) as criteria.**

| Trait | Methionine source | Model | Requirement | $R^2$ | RMSE |
|---|---|---|---|---|---|
| G | DL-Met | QBL | **0.578** | 0.937 | 0.243 |
| | Nano-Met | LALD | 0.530 | 0.896 | 0.312 |
| FCR | DL-Met | QDQA | 0.465 | 0.998 | 0.002 |
| | Nano-Met | QDLA | 0.403 | 0.981 | 0.007 |
| BMY | DL-Met | LBL | 0.569 | 0.973 | 0.227 |
| | Nano-Met | QALD | 0.507 | 0.941 | 0.381 |
| TMY | DL-Met | QAQD | 0.544 | 0.966 | 0.119 |
| | Nano-Met | QAQD | **0.561** | 0.936 | 0.166 |
| MDA | DL-Met | QDQA | 0.572 | 0.999 | 0.026 |
| | Nano-Met | QDQA | 0.544 | 0.958 | 0.277 |

G: body weight gain; FCR: feed conversion ratio; BMY: breast meat yield; TMY: thigh meat yield, MDA: malondialdehyde, LBL: one-slope linear ascending; QBL: one-slope quadratic ascending; LALD: two-slope linear-ascending linear-descending; QDQA: two-slope quadratic-descending quadratic-ascending; QDLA: two-slope quadratic-descending linear ascending; QALD: two-slope quadratic-descending linear-descending; QAQD: two-slope quadratic-ascending quadratic-descending. The highest estimates for DL-Met and Nano-Met were highlighted in boldface.

estimated values of the one-slope with linear ascending portion (LBL) and LALD were statistically different from the commercial recommendations of Aviagen [22].

## Methionine requirements for FCR

Methionine requirements of DL-Met and nano-Met for minimum FCR were estimated at 0.427 and 0.410, 0.472 and 0.435, 0.423 and 0.452, 0.464 and 0.403 and 0.465 and 0.417% of diet using different regression models (Table 4; Fig 4). Two-slope quadratic-descending quadratic-ascending portions (QDQA: $R^2$ = 0.998, RMSE = 0.002) and two-slope quadratic-descending linear-ascending (QDLA: $R^2$ = 0.981, RMSE = 0.007) gave the best estimations, 0.465 and 0.403% for DL-Met and nano-Met, respectively, based on the highest $R^2$ and lowest RMSE. Considering CIs, all estimated values for FCR were statistically different from the commercial recommendations of Aviagen [22].

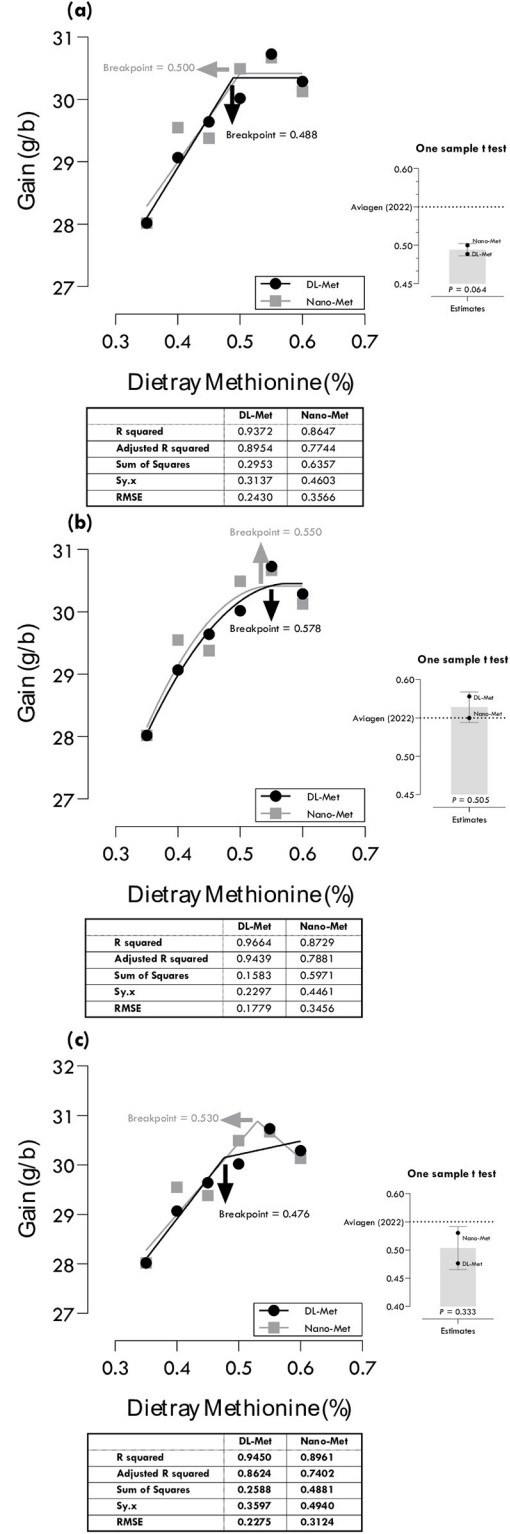

**Fig 3.** Estimation of methionine requirements based on different broken-line models for maximum body weight gain (G): one-slope broken-line ascending-linear (a), one-slope broken-line ascending-quadratic (b), two-slope broken-line ascending-linear descending-linear (c).

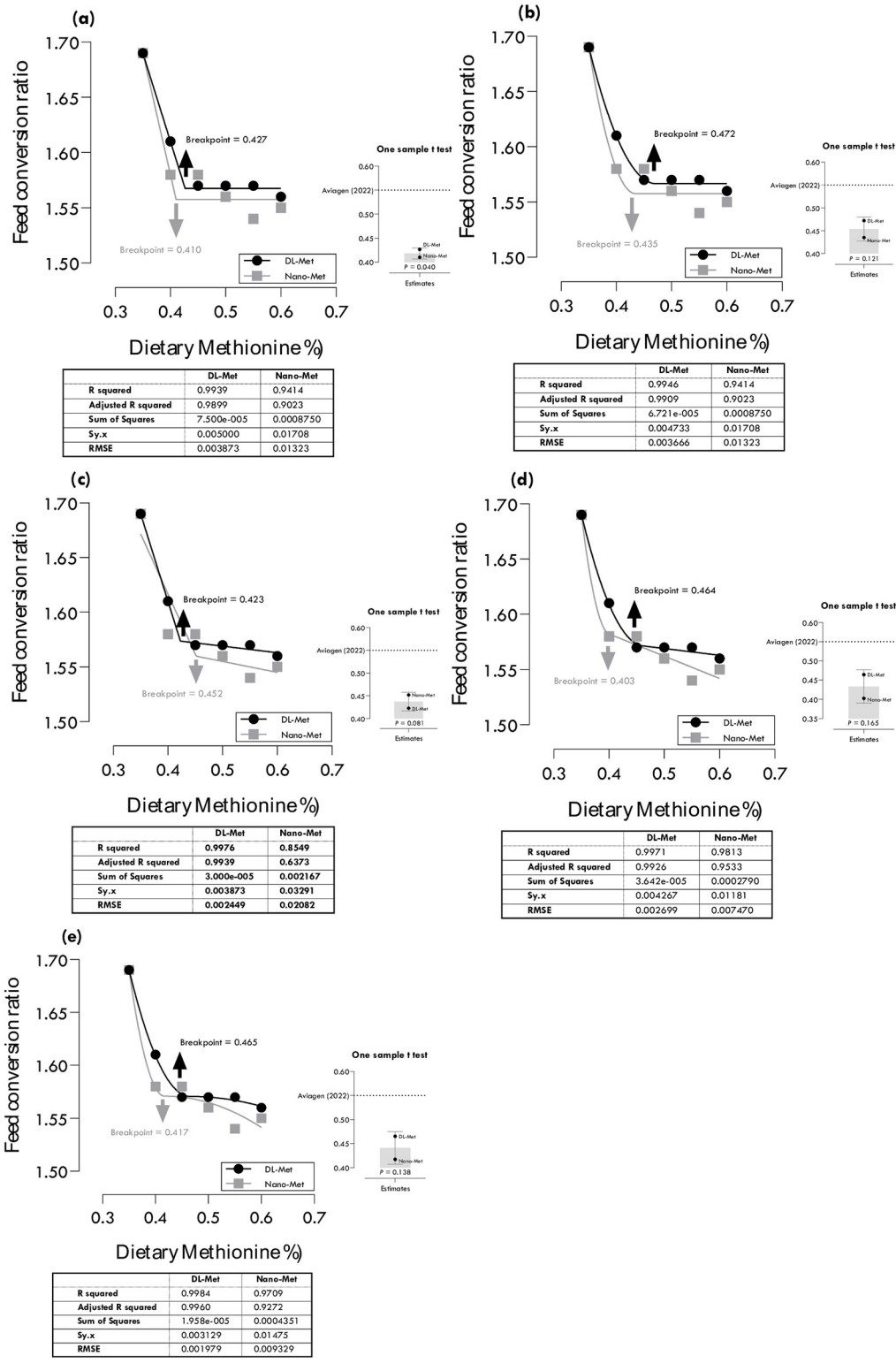

**Fig 4.** Estimation of methionine requirements based on different broken-line models for minimum feed conversion ratio (FCR): one-slope broken-line descending-linear (a), one-slope broken-line descending-quadratic (b), two-slope broken-line descending-linear ascending-linear (c), two-slope broken-line descending-quadratic ascending-linear (d), two-slope broken-line descending-quadratic ascending-quadratic (e).

### Methionine requirements for BMY

Methionine requirements of DL-Met and nano-Met for maximum BMY were estimated at 0.569 and 0.570, 0.418 and 0.523, and 0.480 and 0.507% of diet using different regression models (Table 4; Fig 5). One-slope broken line with linear ascending portion (LBL: $R^2$ = 0.973, RMSE = 0.227) and two-slope quadratic-ascending linear-descending (QALD: $R^2$ = 0.941, RMSE = 0.381) gave the best estimations, 0.569 and 0.507% for DL-Met and nano-Met, respectively, based on the highest $R^2$ and lowest RMSE. Considering confidence intervals (CIs), the estimated values of the LALD and QALD were statistically different from the commercial recommendations of Aviagen [22].

### Methionine requirements for TMY

Methionine requirements of DL-Met and nano-Met for maximum TMY were estimated at 0.438 and 0.423, 0.486 and 0.490, 0.531 and 0.540, 0.498 and 0.450, and 0.544 and 0.561% of diet using different regression models (Table 4; Fig 6). Two-slope quadratic-ascending quadratic-descending portion (QAQD: $R^2$ = 0.966, RMSE = 0.119) and QAQD ($R^2$ = 0.936, RMSE = 0.166) gave the best estimations, 0.544 and 0.561% for DL-Met and nano-Met, respectively, based on the highest $R^2$ and lowest RMSE. Considering CIs, all estimated values for FCR were statistically different from the commercial recommendations of Aviagen [22].

### Methionine requirements for MDA

Methionine requirements of DL-Met and nano-Met for minimum MDA were estimated at 0.508 and 0.490, 0.572 and 0.547, 0.529 and 0.487, 0.572 and 0.543, and 0.572 and 0.544% of diet using different regression models (Table 4; Fig 7). The QDQA ($R^2$ = 0.999, RMSE = 0.026 for DL-Met and $R^2$ = 0.958, RMSE = 0.277 for nano-Met) gave the best estimations, 0.572 and 0.544% for DL-Met and nano-Met, respectively, based on the highest $R^2$ and lowest RMSE. Considering CIs, the estimated values by LBL and LALD models for minimum MDA were statistically different from the commercial recommendations of Aviagen [22].

## Discussion

Based on the results, it is clear that supplementation of methionine sources had a significant positive impact on the growth, feed efficiency, meat quality, and carcass yield of broilers. The increase in growth performance and feed efficiency indicates that the supplemental methionine sources were able to improve the performance of the broilers and nano-Met was tested within a sensitive range. The highest effect size of supplemental methionine was on MDA ($\eta^2 p$ = 0.924), followed by FCR ($\eta^2 p$ = 0.578), BMY ($\eta^2 p$ = 0.575), BWG ($\eta^2 p$ = 0.430), and TMY ($\eta^2 p$ = 0.332), suggesting the potent antioxidant properties of methionine. Dietary methionine plays a crucial role in the antioxidant system of chickens. Research has shown that methionine supplementation above the requirement can improve the antioxidant status and reduce oxidative stress in poultry [29]. Methionine is involved in the synthesis of other sulfur amino acids, such as cysteine, which is important for the production of glutathione, a key antioxidant in the body [30]. Additionally, methionine supplementation has been linked to mitigating intestinal oxidative stress and enhancing the immune function of poultry, further emphasizing its role in the antioxidant defense system [31]. Therefore, adequate levels of dietary methionine are essential for maintaining the antioxidant balance and overall health of chickens [32].

Previous studies have produced variable results when it comes to the bioavailability of different methionine sources in broilers [4, 7]. Some studies have found that the methionine

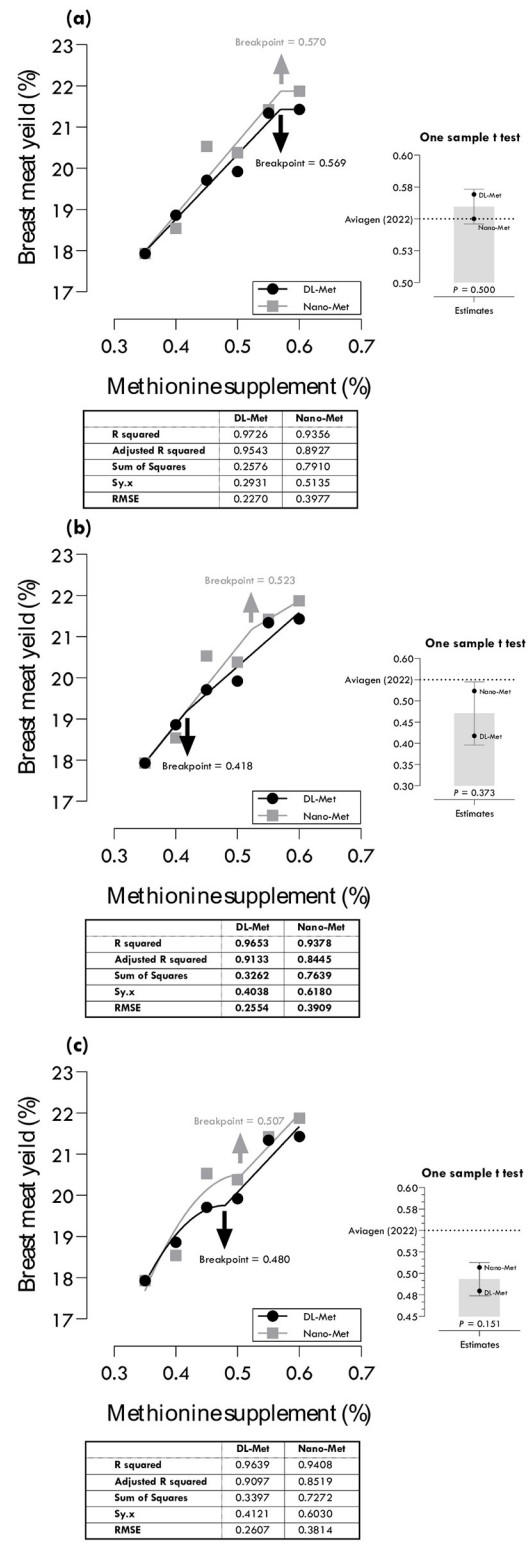

**Fig 5.** Estimation of methionine requirements based on different broken-line models for maximum breast meat yield (BMY): one-slope broken-line ascending-linear (a), two-slope broken-line ascending-linear descending-linear (b), two-slope broken-line ascending-quadratic descending-linear (c).

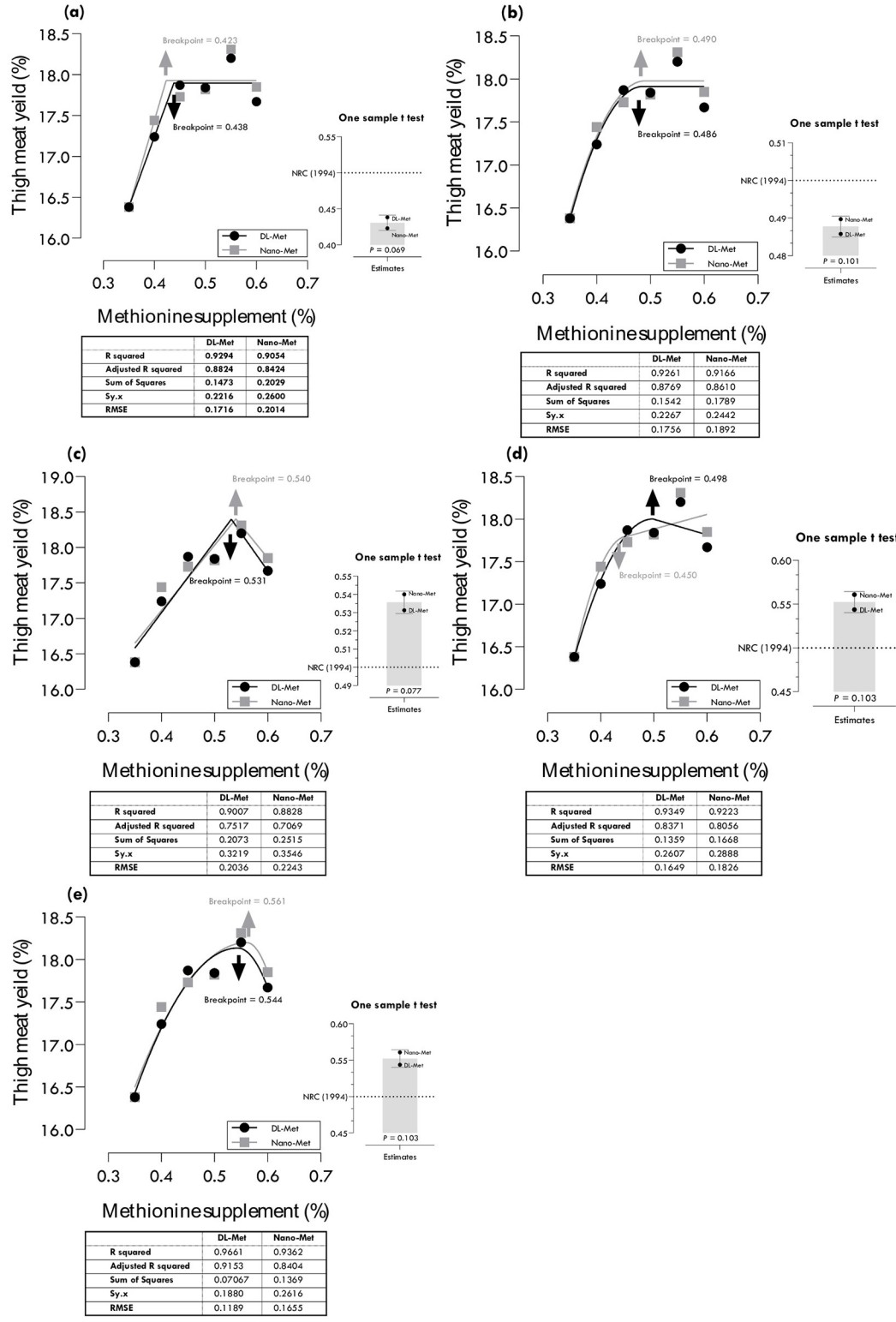

**Fig 6.** Estimation of methionine requirements based on different broken-line models for maximum thigh meat yield (TMY): one-slope broken-line ascending-linear (a), one-slope broken-line ascending-quadratic (b), two-slope broken-line ascending-linear descending-linear (c), two-slope broken-line ascending-quadratic descending-linear (d), two-slope broken-line ascending-quadratic descending-quadratic (e).

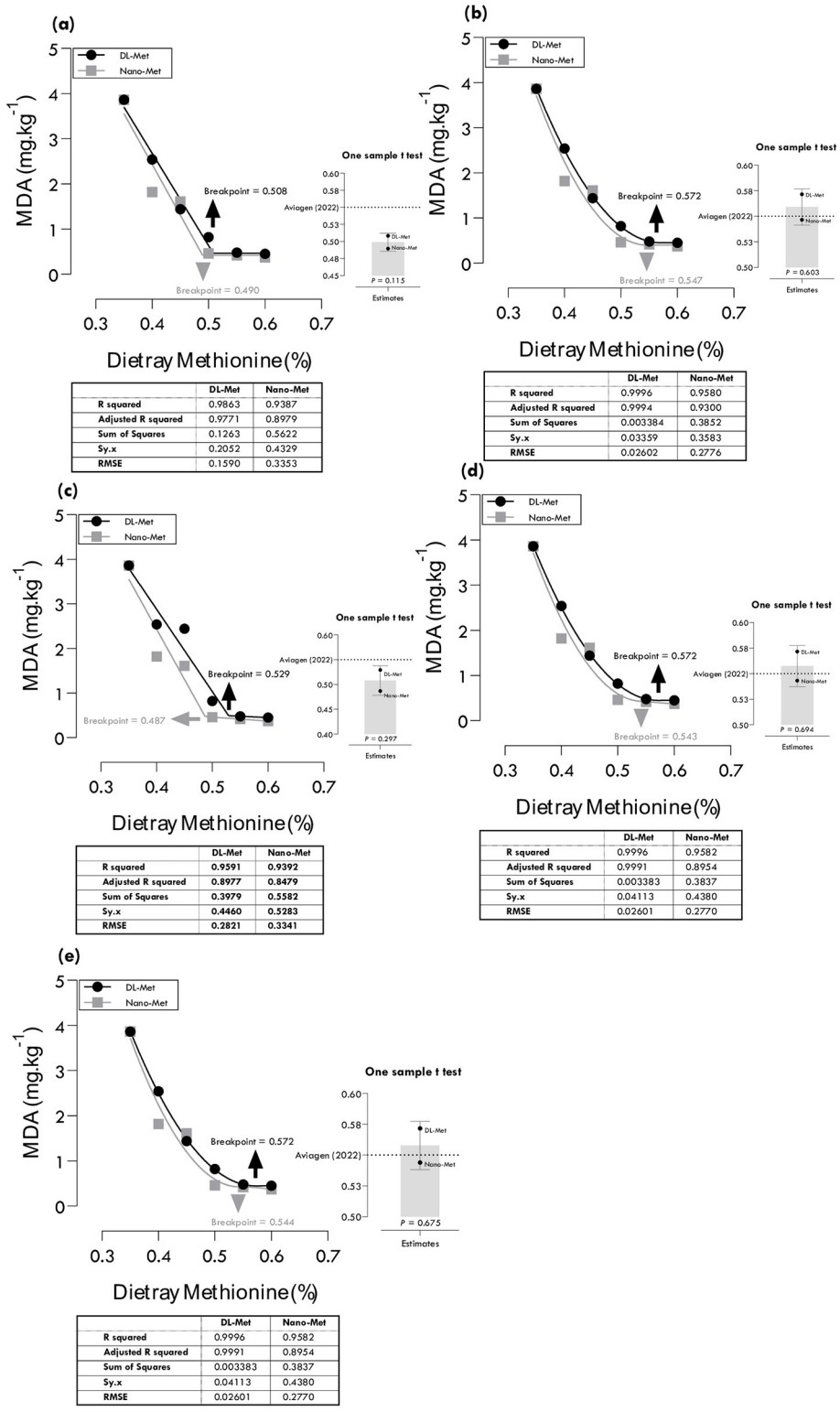

**Fig 7.** Estimation of methionine requirements based on different broken-line models for minimum malondialdehyde (MDA) production in meat samples: one-slope broken-line ascending-linear (a), one-slope broken-line ascending-quadratic (b), two-slope broken-line ascending-linear descending-linear (c), two-slope broken-line ascending-quadratic descending-linear (d), two-slope broken-line ascending-quadratic descending-quadratic (e).

source does not significantly affect broiler performance or carcass yield [5]. However, other studies have shown that the methionine source can have an impact on blood characteristics, antioxidant response, and growth performance [33, 34].

The methionine nanoparticles used for this study were produced using an ultrasonic-assistance technique, which physically reduced the size of the methionine granules down to the nanoscale level without altering their chemical structure. In this technique, the methionine granules were milled to the desired particle size using high-intensity ultrasonic waves. These waves create mechanical stress, which results in the physical breakage of the granules, leading to the formation of the nanoparticles. The nanoparticles were confirmed under an electron microscope (Fig 1a), which showed that the particle size was in the nanoscale range (average particle size ≈ 50 nm). No chemical alteration occurred during this process, indicating that the nanoparticles are chemically similar to the original methionine used (Fig 1b). Our research specifically focused on comparing the bioavailability of nano-Met and DL-Met in broilers. Nano-Met refers to methionine that has been processed into nanoparticles, which can enhance its absorption and utilization in the body. DL-Met, on the other hand, is a commonly used synthetic form of methionine in poultry diets. The results of our study indicated that nano-Met had a higher RBV compared to DL-Met, however, these differences were not statistically significant. Table 3 shows that although there was a significant difference in the slope for TMY, the reference value of the standard (DL-Met = 100%) fell within the confidence interval range of 27–193% (Fig 2), resulting in the non-significance of RBV of nano-Met relative to DL-Met. This indicates that the difference between the two values is not statistically significant, and the confidence interval suggests that the true value of RBV could be anywhere within this range [27]. On the contrary, the numerical increase in RBV of nano-Met implies that it may be more effectively absorbed and utilized by broilers, leading to improved growth and performance. This increased bioavailability of nano-Met can be attributed to its improved solubility and absorption properties due to its nanoparticle form. This suggests that the utilization of nano-Met could have positive implications for broiler health and productivity.

Overall, nanotechnology has the potential to increase the RBV of nanoparticles, leading to improved performance and outcomes in various fields. By optimizing the size, shape, surface properties, and delivery systems of nanoparticles, their bioavailability can be enhanced, which can have significant implications for human and animal health, agriculture, and food science. However, further research is needed to fully understand the mechanisms and optimal conditions for increasing the RBV of nanoparticles such as nano-Met.

The advantages of using different models to estimate nutrient requirements include improved performance, accurate estimation, easy implementation, and cautious estimation. More complex models with multiple nutritional inputs may improve performance in commercial applications [19]. Different models allow for more accurate estimation of nutrient requirements, which is crucial for optimizing responses without moving into inhibiting or toxic ranges of nutrients [18, 20, 21, 35]. The present study presented various models for determining the nutritional requirement of methionine, which varied in accuracy for different responses. For example, the one-slope broken-line model demonstrated the highest accuracy for estimating the methionine requirement for maximum BWG, while the two-slope broken-line model was the most suitable for estimating the methionine requirement for maximum TMY and minimum MDA. Therefore, the choice of model should be based on the best estimation for a particular response. Combining the advantages of different models can result in models that are easy to implement and more accurately estimate nutrient requirements [36]. Some models provide a cautious estimate of nutritional requirements, which can be beneficial for ensuring that the requirements are not underestimated [19]. The current study demonstrated that different models may produce different estimated values for the requirement of

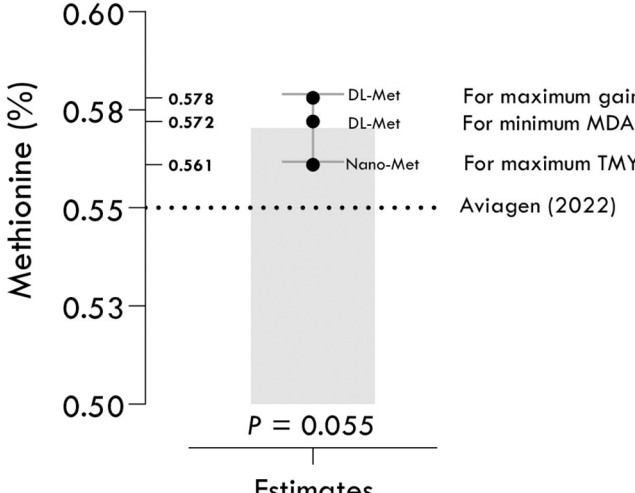

**Fig 8. Comparison of the maximum estimates for DL-methionine (DL-Met) and methionine nanoparticles (nano-Met) with the recommended methionine requirement by NRC (1994).**

methionine for chickens based on one sample t-test (Fig 8). If the confidence interval does not include the theoretical value in a one-sample t-test, it means that the estimated values are significantly different from the theoretical value. This is because the confidence interval represents a range of values that are likely to contain the true population mean with a certain level of confidence, and if the theoretical value is not within this range, it is unlikely to be the true population mean. The fact that the P-values were not significant does not affect this conclusion [37, 38].

## Conclusion

The findings of our research have important implications for the poultry industry. By using nano-Met as a methionine source in broiler diets, producers can potentially improve the efficiency of nutrient utilization and enhance broiler performance. This can lead to better growth rates, carcass yield, and overall profitability. It is worth noting that further research is needed to fully understand the mechanisms behind the higher bioavailability of nano-Met and to determine the optimal inclusion levels in broiler diets. Additionally, studies comparing nano-Met to other methionine sources, such as L-Met, could provide valuable insights into the most effective and cost-efficient options for broiler nutrition. This study also emphasizes the significance of implementing various regression models for estimating the nutritional requirements of nutrients in the feeding of broiler chickens.

## Supporting information

**S1 Data.**
(RAR)

## Acknowledgments

The authors would like to acknowledge the help and guidance provided by Dr. Ranjbar and Mr. Hamid-Reza Vorrasi in performing the methionine nanoparticle synthesis process.

## Author Contributions

**Conceptualization:** Mehran Mehri.

**Data curation:** Mahmoud Ghazaghi, Mehravar Mehri, Morteza Asghari-Moghadam, Mehran Mehri.

**Formal analysis:** Mehran Mehri.

**Funding acquisition:** Morteza Asghari-Moghadam, Mehran Mehri.

**Investigation:** Mahmoud Ghazaghi, Mehravar Mehri, Morteza Asghari-Moghadam, Mehran Mehri.

**Methodology:** Mehravar Mehri, Mehran Mehri.

**Project administration:** Mahmoud Ghazaghi, Mehravar Mehri, Morteza Asghari-Moghadam, Mehran Mehri.

**Resources:** Mahmoud Ghazaghi, Morteza Asghari-Moghadam, Mehran Mehri.

**Software:** Mehran Mehri.

**Supervision:** Morteza Asghari-Moghadam, Mehran Mehri.

**Validation:** Mahmoud Ghazaghi, Mehravar Mehri, Mehran Mehri.

**Visualization:** Mahmoud Ghazaghi, Morteza Asghari-Moghadam, Mehran Mehri.

**Writing – original draft:** Mahmoud Ghazaghi, Mehravar Mehri, Morteza Asghari-Moghadam, Mehran Mehri.

**Writing – review & editing:** Mahmoud Ghazaghi, Mehravar Mehri, Morteza Asghari-Moghadam, Mehran Mehri.

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
