## [Decision Letter · Decision Letter 0]

4 Mar 2024

PONE-D-23-43596A novel methionine nanoparticle in broiler chickens: Bioavailability and requirementsPLOS ONE

Dear Dr. Mehri,

Thank you for submitting your manuscript to PLOS ONE. After careful consideration, we feel that it has merit but does not fully meet PLOS ONE’s publication criteria as it currently stands. Therefore, we invite you to submit a revised version of the manuscript that addresses the points raised during the review process.

We look forward to receiving your revised manuscript.

Kind regards,

Mohammed Fouad El Basuini, Professor

Academic Editor

PLOS ONE

Journal Requirements:

Reviewers' comments:

Reviewer's Responses to Questions

**Comments to the Author**

1. Is the manuscript technically sound, and do the data support the conclusions?

Reviewer #1: Partly

Reviewer #2: Yes

2. Has the statistical analysis been performed appropriately and rigorously? 

Reviewer #1: Yes

Reviewer #2: No

3. Have the authors made all data underlying the findings in their manuscript fully available?

Reviewer #1: Yes

Reviewer #2: No

4. Is the manuscript presented in an intelligible fashion and written in standard English?

Reviewer #1: Yes

Reviewer #2: Yes

5. Review Comments to the Author

Reviewer #1: Comments to the Authors

I have conducted a thorough examination of the manuscript, and it is evident that substantial revisions are needed. Below are some of my comments:

Comment #1. Line 23, 24. "p < 0.05" should be italicized. Pleasse check the whole manscript.

Comment #2. Abstract. In conclusion, it would be beneficial to include information regarding the optimal dose of methionine supplementation. Please consider adding this detail.

Comment #3: Keywords. It would be beneficial to enrich the study by including important keywords related to the research. Consider incorporating specific terms such as the name of the animal involved and the substance used.

Comment #4. Introduction. Consider emphasizing the research problem and the novelty more towards the end of the introduction for better clarity. Please rewrite it clearly with mention the specific parameters to achieve the aim.

Comment #5: Line 145. Preparation of Methionine Nanoparticles. It would be beneficial to provide references for the methodology used in the preparation of methionine nanoparticles. This helps readers and researchers to access additional information, ensuring transparency and credibility in the study.

Comment #6: Line 207 and 208. Inconsistent Figure References. Please use either the abbreviation 'Fig.' or the full name 'Figure' consistently for references such as 'Fig. 1a' and 'Figure 2b' to maintain uniformity throughout the document. Please check the entire manuscript.

Comment #7: Results. In the section spanning from Line 206 to 212, it would be beneficial to include a descriptive title for the description of this result. A clear and concise title will provide readers with a preview of the content, aiding in the overall organization and comprehension of the manuscript.

Comment #8: Results. In Line 213, the term "Analysis of variance" is used as a title. Please consider if this title accurately reflects the specific content and findings presented in this section. If possible, provide a more specific and descriptive title that aligns with the nature of the results discussed, ensuring clarity for readers.

Comment #9: Line 247, 255, 262, 269, 276, etc. Inconsistency in Reference Format. The citation 'Aviagen (2022)' appears multiple times throughout the manuscript. Please clarify whether 'Aviagen (2022)' is a report, a publication, or another type of reference.

Comment #10: Conclusion. It is recommended to place the 'Conclusion' section under a separate heading for improved organization and readability. This would enhance the structure of the manuscript and make it easier for readers to navigate and comprehend the key findings and implications.

Comment #11: Tables. The manuscript makes references to several tables, but they appear to be missing. Please ensure that the tables mentioned in the text are included in the manuscript.

Comment #12: Figures. It is observed that all the figures in the manuscript are without captions. Please check it.

Comment #13. I have noticed a few minor errors and spelling mistakes scattered throughout the manuscript. To enhance the linguistic quality of the manuscript, I would recommend considering professional language revision.

Reviewer #2: I am pleased to participate in reviewing this interesting research manuscript. The study reported a patented formulation of DL methionine into nanoparticles and investigated the positive effect of DL methionine supplementation on broiler performance. The study also statistically compared the performance parameters in response to DL methionine forms “powder and nanoparticles” supplementation using well-designed statistical models. However, I believe that the manuscript needs a major revision to reduce the redundancy and fulfill the following comments.

- The standard performance parameters (body weight and FCR) should be mentioned at least to know how far the results are, when compared to the standard.

- I admire your statistical proficiency. Therefore, I would recommend adding a power analysis for the used sample size.

- Figures’ captions should be provided. The caption should include detailed information about the figure that can be stand-alone to understand the figure without reading the manuscript.

- Where are the Tables (1 – 3), I could not find them in the manuscript?!!

- Line (L) 19: “….to create 11 experimental diets including a basal diet and…” it is confusing. Please add a comma “,” before “including”

- L 43: Reference [3] seems irrelevant to the reported information. Please cite a more relevant reference.

- L 145 – 153: What was the source of lipid used in the methionine nanoparticle formation?

- L 234: Remove “FCR BMY”.

- L (322 – 343): “Nanotechnology has been increasingly ……..” Redundant information and unnecessary details should be omitted. It is already explained in the introduction. Please remove the whole paragraph L (322 – 343).

- Five statistical models were used to analyze FCR, TMY, and MDA (Fig 4, 6, and 7), while only three statistical models were used to analyze BWG and BMY (Fig 3 and 5). I believe that you should be consistent. Please explain if you have a rationale for not being consistent.

- Where is the “Conclusion”?! I guess that it is the last paragraph. Either you call the “Discussion” section (Discussion and conclusion), or you add the section title “Conclusion” before the last paragraph.

- Fig. 8 has not been mentioned in the manuscript. Either you discuss the figure or remove it.

- The supplementary data also has not been mentioned in the manuscript. Further, the “CSV” tables contain values with column titles “Met, BRST, and THIGH”. Should the reader guess?!!!

6. PLOS authors have the option to publish the peer review history of their article (what does this mean?). If published, this will include your full peer review and any attached files.

Reviewer #1: No

Reviewer #2: **Yes: **Ahmed F A Ghareeb

---

## [Author Response · Author response to Decision Letter 0]

9 Mar 2024

RESPONSE TO REVIEWERS

Dear Editor,

I would like to express my gratitude to the respected reviewers who evaluated my manuscript for the PLOS one. I have amended the manuscript according to the comments of the respected reviewers, which are highlighted in the text for easy tracking.

Sincerely,

Mehran Mehri (Corresponding author) 

Reviewer #1

I have conducted a thorough examination of the manuscript, and it is evident that substantial revisions are needed. Below are some of my comments:

Comment #1. Line 23, 24. "p < 0.05" should be italicized. Pleasse check the whole manscript.

AU: Done.

Comment #2. Abstract. In conclusion, it would be beneficial to include information regarding the optimal dose of methionine supplementation. Please consider adding this detail.

AU: The requested information was already included in the ABSTRACT. Please refer to L28.

Comment #3: Keywords. It would be beneficial to enrich the study by including important keywords related to the research. Consider incorporating specific terms such as the name of the animal involved and the substance used.

AU: Done.

Comment #4. Introduction. Consider emphasizing the research problem and the novelty more towards the end of the introduction for better clarity. Please rewrite it clearly with mention the specific parameters to achieve the aim.

AU: Thank you for your suggestion. Done.

Comment #5: Line 145. Preparation of Methionine Nanoparticles. It would be beneficial to provide references for the methodology used in the preparation of methionine nanoparticles. This helps readers and researchers to access additional information, ensuring transparency and credibility in the study.

AU: Thank you for your suggestion. Done.

Comment #6: Line 207 and 208. Inconsistent Figure References. Please use either the abbreviation 'Fig.' or the full name 'Figure' consistently for references such as 'Fig. 1a' and 'Figure 2b' to maintain uniformity throughout the document. Please check the entire manuscript. 

AU: Done.

Comment #7: Results. In the section spanning from Line 206 to 212, it would be beneficial to include a descriptive title for the description of this result. A clear and concise title will provide readers with a preview of the content, aiding in the overall organization and comprehension of the manuscript.

AU: Thank you for your suggestion. Done.

Comment #8: Results. In Line 213, the term "Analysis of variance" is used as a title. Please consider if this title accurately reflects the specific content and findings presented in this section. If possible, provide a more specific and descriptive title that aligns with the nature of the results discussed, ensuring clarity for readers.

AU: Thank you for your suggestion. Done.

Comment #9: Line 247, 255, 262, 269, 276, etc. Inconsistency in Reference Format. The citation 'Aviagen (2022)' appears multiple times throughout the manuscript. Please clarify whether 'Aviagen (2022)' is a report, a publication, or another type of reference.

AU: Thank you for your suggestion. Done.

Comment #10: Conclusion. It is recommended to place the 'Conclusion' section under a separate heading for improved organization and readability. This would enhance the structure of the manuscript and make it easier for readers to navigate and comprehend the key findings and implications.

AU: Done.

Comment #11: Tables. The manuscript makes references to several tables, but they appear to be missing. Please ensure that the tables mentioned in the text are included in the manuscript.

AU: Thank you for your suggestion. Done.

Comment #12: Figures. It is observed that all the figures in the manuscript are without captions. Please check it.

AU: Thank you for your suggestion. Done.

Comment #13. I have noticed a few minor errors and spelling mistakes scattered throughout the manuscript. To enhance the linguistic quality of the manuscript, I would recommend considering professional language revision.

AU: Thank you for your suggestion. Done.

Reviewer #2

I am pleased to participate in reviewing this interesting research manuscript. The study reported a patented formulation of DL methionine into nanoparticles and investigated the positive effect of DL methionine supplementation on broiler performance. The study also statistically compared the performance parameters in response to DL methionine forms “powder and nanoparticles” supplementation using well-designed statistical models. However, I believe that the manuscript needs a major revision to reduce the redundancy and fulfill the following comments.

- The standard performance parameters (body weight and FCR) should be mentioned at least to know how far the results are, when compared to the standard.

AU: Thank you for bringing this to my attention. I understand your suggestion, which implies including the standard performance data of broilers reported by Aviagen. However, it's worth noting that these reported performances were achieved under optimal commercial conditions, resulting in inevitably higher values compared to those observed under our research conditions. Consequently, we opted not to utilize these data in our report.

- I admire your statistical proficiency. Therefore, I would recommend adding a power analysis for the used sample size.

AU: Thank you for your suggestion. Done.

- Figures’ captions should be provided. The caption should include detailed information about the figure that can be stand-alone to understand the figure without reading the manuscript.

AU: Thank you for your suggestion. Done.

- Where are the Tables (1 – 3), I could not find them in the manuscript?!!

AU: Thank you for your suggestion. The Tables were included.

- Line (L) 19: “….to create 11 experimental diets including a basal diet and…” it is confusing. Please add a comma “,” before “including”

AU: Thank you for your suggestion. Done.

- L 43: Reference [3] seems irrelevant to the reported information. Please cite a more relevant reference.

AU: Thank you for your notice, replaced.

- L 145 – 153: What was the source of lipid used in the methionine nanoparticle formation?

AU: As described before, the liquid was the solution of the DL-Met and surfactant.

- L 234: Remove “FCR BMY”.

AU: Thank you for your notice, removed.

- L (322 – 343): “Nanotechnology has been increasingly ……..” Redundant information and unnecessary details should be omitted. It is already explained in the introduction. Please remove the whole paragraph L (322 – 343).

AU: Thank you for your suggestion, removed.

- Five statistical models were used to analyze FCR, TMY, and MDA (Fig 4, 6, and 7), while only three statistical models were used to analyze BWG and BMY (Fig 3 and 5). I believe that you should be consistent. Please explain if you have a rationale for not being consistent.

AU: We used five models but in some cases some models failed to converge.

- Where is the “Conclusion”?! I guess that it is the last paragraph. Either you call the “Discussion” section (Discussion and conclusion), or you add the section title “Conclusion” before the last paragraph.

AU: Thank you for your suggestion, included.

- Fig. 8 has not been mentioned in the manuscript. Either you discuss the figure or remove it.

AU: Please refer to L350.

- The supplementary data also has not been mentioned in the manuscript. Further, the “CSV” tables contain values with column titles “Met, BRST, and THIGH”. Should the reader guess?!!!

AU: Done.

---

## [Decision Letter · Decision Letter 1]

1 Apr 2024

A novel methionine nanoparticle in broiler chickens: Bioavailability and requirements

PONE-D-23-43596R1

Dear Dr. Mehri,

We’re pleased to inform you that your manuscript has been judged scientifically suitable for publication and will be formally accepted for publication once it meets all outstanding technical requirements.

Kind regards,

Mohammed Fouad El Basuini, Professor

Academic Editor

PLOS ONE

Additional Editor Comments (optional):

Reviewers' comments:

Reviewer's Responses to Questions

**Comments to the Author**

1. If the authors have adequately addressed your comments raised in a previous round of review and you feel that this manuscript is now acceptable for publication, you may indicate that here to bypass the “Comments to the Author” section, enter your conflict of interest statement in the “Confidential to Editor” section, and submit your "Accept" recommendation.

Reviewer #1: All comments have been addressed

Reviewer #2: All comments have been addressed

2. Is the manuscript technically sound, and do the data support the conclusions?

Reviewer #1: Yes

Reviewer #2: Yes

3. Has the statistical analysis been performed appropriately and rigorously? 

Reviewer #1: Yes

Reviewer #2: Yes

4. Have the authors made all data underlying the findings in their manuscript fully available?

Reviewer #1: Yes

Reviewer #2: Yes

5. Is the manuscript presented in an intelligible fashion and written in standard English?

Reviewer #1: Yes

Reviewer #2: Yes

6. Review Comments to the Author

Reviewer #1: After thorough review and consideration, I recommend accepting this new version of the manuscript for publication.

Reviewer #2: As far as I concerned, the author response fulfilled the comments.

I do not have further comments or concerns.

7. PLOS authors have the option to publish the peer review history of their article (what does this mean?). If published, this will include your full peer review and any attached files.

Reviewer #1: No

Reviewer #2: No

---

## [Editor Report · Acceptance letter]

4 Apr 2024

PONE-D-23-43596R1 

PLOS ONE

Dear Dr. Mehri, 

I'm pleased to inform you that your manuscript has been deemed suitable for publication in PLOS ONE. Congratulations! Your manuscript is now being handed over to our production team.

Kind regards, 

on behalf of

Dr Mohammed Fouad El Basuini 

Academic Editor

PLOS ONE